# Food Hypersensitivity: Distinguishing Allergy from Intolerance, Main Characteristics, and Symptoms—A Narrative Review

**DOI:** 10.3390/nu17081359

**Published:** 2025-04-16

**Authors:** Gregory Hage, Yonna Sacre, Joanne Haddad, Marcel Hajj, Lea Nicole Sayegh, Nicole Fakhoury-Sayegh

**Affiliations:** 1Department of Nutrition and Food Science, Faculty of Arts and Sciences, Holy Spirit University of Kaslik, Jounieh P.O. Box 446, Lebanon; 2Hajj Medical Center-Medical & Dental Clinics, Green Zone A Building 71 Ground Floor, Naccache P.O. Box 1201, Lebanon; 3Faculty of Dental Medicine, Saint Joseph University of Beirut, Medical Sciences Campus, Damascus Road, Riad Solh, Beirut P.O. Box 11-5076, Lebanon; 4Yale New Haven Hospital, P.O. Box 1880, 20 York Street, New Haven, CT 06510, USA; 5Faculty of Pharmacy, Department of Nutrition, Saint Joseph University of Beirut, Medical Sciences Campus, Damascus Road, Riad Solh, Beirut P.O. Box 11-5076, Lebanon

**Keywords:** food hypersensitivity, food allergy, food intolerance, clinical manifestations, IBS

## Abstract

**Background/Objectives:** Food hypersensitivity remains an understudied and overlooked subject globally. It is characterized by adverse reactions to dietary substances, potentially triggered by various mechanisms. Food allergy, a subset of food hypersensitivity, denotes an immune response to food proteins categorized into immunoglobulin IgE-mediated or non-IgE-mediated reactions. Conversely, food intolerance, another facet of food hypersensitivity, refers to non-immunological reactions, in which the human body cannot properly digest certain foods or components, leading to gastrointestinal discomfort and other non-immune-related symptoms. The main objective of this study was to determine and differentiate the differences, characteristics, and types of food hypersensitivity. **Methods**: This study involved a comprehensive review of key research from 1990 onward, including review articles, prospective studies, nested case–control studies, and meta-analyses. **Results:** Recognizing these differences is essential for healthcare professionals to ensure accurate diagnosis, effective management, and improved patient outcomes, while also aiding dietitians in providing optimal nutritional and dietary guidance. **Conclusions:** there are big differences between the main characteristics, such as symptoms, complications, and treatments between allergies, and food intolerances. Commonly reported trigger foods include cow milk, gluten, eggs, nuts, and seafood.

## 1. Introduction

Food hypersensitivity is a prevalent condition, affecting individuals worldwide, characterized by adverse reactions to specific food components [1]. It encompasses both food allergies and immune-mediated reactions and intolerances (non-immune-mediated reactions), which can manifest with a variety of symptoms, ranging from mild discomfort to severe life-threatening reactions [1] such as skin manifestations (rash, dermatitis, rosacea, and angioedema), gastrointestinal symptoms (bloating, diarrhea, constipation, gas, reflux, and macro- and micronutrient malabsorption), and respiratory symptoms (anaphylactic shock, shortness of breath, nasal congestion, rhinorrhea, sneezing, itching of the nose and throat, coughing, and wheezing) [1].

The prevalence of food hypersensitivity has been steadily increasing, with food allergies affecting approximately 2–10% of the population, a number varying by region and age group [2]. Studies indicate that immunoglobulin E (IgE)-mediated food allergies are the most common in children, whereas non-IgE-mediated hypersensitivities and intolerances are more frequently observed in adults [3]. Lactose intolerance, for instance, is a well-documented form of non-immune-mediated hypersensitivity, affecting over 65% of the global population due to lactase enzyme deficiency [4]. Similarly, non-celiac gluten sensitivity has gained recognition as a distinct condition, though its pathophysiology remains under investigation [5].

Distinguishing between food allergy and intolerance is crucial for appropriate management and treatment strategies [6] (Figure 1). This narrative review aimed to elucidate the differences between food allergies and intolerances, including their epidemiology, pathophysiology, clinical manifestations, diagnostic approaches, and management strategies. By understanding the nuances of these conditions, healthcare professionals can provide better care, treatment, prevention, and nutritional management and improve the quality of life of individuals suffering from food hypersensitivities.

## 2. Material and Methods

This research utilized the Medline database, accessed via its search engine, PubMed, and Google Scholar. The following Medical Subject Heading (MeSH) terms or text words were used: “Food hypersensitivity” or “food allergies” or “Food intolerances”

The studies included in this review focused on human subjects, were published in English, and had publication dates from 1990 onwards. Specifically, we prioritized prospective studies, meta-analyses, systematic reviews, narrative reviews, and nested case–control studies.

We excluded studies published before 1990, animal studies, and non-English language articles.

For precision purposes, two researchers reviewed all relevant articles independently to check for discrepancies and ensure rigorous data extraction. For each article, we focused on the key study characteristics, including publication year, country of origin, study design, sample size, and participant characteristics.

One thousand five hundred and thirty-four records were identified through database searches. After removing 321 duplicates, 1213 records remained for screening. Of these, 954 records were excluded for not meeting inclusion criteria, leaving 259 full-text articles for eligibility assessment. This resulted in 142 studies being included in the final narrative synthesis.

Details of the study selection process are illustrated in Figure 2.

The review is divided into 3 sections, highlighting the 2 main components of food hypersensitivity, which are food allergies and food intolerance, with the 3rd part being about the relation of food hypersensitivity with IBS and the role of microbiota. Some of the main characteristics of the studies that met the inclusion criteria can be found in Table 1 [7,8,9,10,11,12,13,14].

## 3. Results

### 3.1. Food Allergies

#### 3.1.1. Definition

A food allergy, categorized under food hypersensitivity, refers to an immunological reaction to food proteins [15]. This reaction can involve immunoglobulin IgE-mediated responses, mixed IgE- and non-IgE-mediated reactions, or non-IgE-mediated responses [4,5]. IgE-mediated food allergies affect millions worldwide, significantly impacting individuals’ daily lives [6]. The prevalence of food allergies is estimated to affect 10–25% of the global population, with an upward trend observed over the last two decades [16]. The classification of food allergies is summarized in Figure 1 [17].

#### 3.1.2. Symptoms

Symptoms may affect various systems including the respiratory tract, the gastrointestinal tract, skin, and the cardiovascular system [18]. Respiratory symptoms encompass sneezing, congestion, rhinorrhea, wheezing, and laryngeal edema [18]. Gastrointestinal manifestations consist of nausea, vomiting, abdominal pain, and diarrhea [18]. Cutaneous presentations may include urticaria, angioedema, flushing, or pruritus [18]. Cardiovascular signs may involve tachycardia, hypotension, or syncope [18]. Typically, symptoms manifest within minutes of food ingestion, although they may be delayed by up to 2 h [18]. The severity ranges from pruritus alone to anaphylactic shock [18]. The symptoms for each food allergy disorder can be found in Table 2 [17].

#### 3.1.3. IgE-Mediated Food Allergies

Regarding IgE-mediated food allergies, they are categorized as type I hypersensitivity reactions. Symptoms typically manifest rapidly, occurring within minutes to a few hours after ingestion. Diagnosis involves a detailed assessment of clinical symptoms and various testing methods, including oral food challenges, food allergy skin prick testing, and specific food IgE testing [19]. Furthermore, IgE-mediated food allergies are characterized by the immune system’s production of IgE antibodies in response to specific food proteins, leading to rapid allergic reactions that can range from mild symptoms to anaphylaxis. Several studies have examined the prevalence and mechanisms of these allergies. One study highlighted that the incidence of IgE-mediated food allergies is increasing globally, particularly among children, with common allergens including peanuts, tree nuts, and shellfish [20]. Another study by Savage et al. compared different diagnostic approaches, such as oral food challenges and specific IgE testing, noting that while specific IgE testing is useful, it can sometimes result in false positives, making oral food challenges the gold standard for diagnosis [21]. Furthermore, research by Nwaru et al. emphasized that the early introduction of allergenic foods might reduce the risk of developing IgE-mediated food allergies, a finding that has influenced recent guidelines on allergy prevention [13]. These studies collectively underscore the complexity of diagnosing and managing IgE-mediated food allergies and the need for tailored approaches based on individual patient profiles.

##### Diagnosis Methods

Skin prick testing (SPT) serves as a common diagnostic tool for identifying type I hypersensitivity reactions. It involves applying an allergen extract onto the skin, typically on the forearm or back, followed by pricking the skin with a lancet [22]. This minimally invasive test offers the advantage of assessing multiple allergens, including inhalants, foods, drugs, venom, latex, and occupational allergens, within a short timeframe of 15–20 min, with a high sensitivity of 95–100% [22].

Skin prick testing (SPT) is a widely used method for diagnosing IgE-mediated allergic conditions by exposing the skin to small amounts of suspected allergens and observing a localized allergic reaction. This method is often compared with other diagnostic approaches like specific IgE (sIgE) testing. A study by Heinzerling et al. found that SPT is highly sensitive, particularly to respiratory allergens such as pollen, mold, and dust mites, and provides rapid results, making it a first-line test in many allergy clinics. However, the study also noted variability in results depending on factors like the allergen extract quality, the technique used, and the patient’s skin condition at the time of testing [22].

Another study by Bousquet et al. explored the correlation between SPT results and clinical symptoms and found a strong correlation between positive SPT results and clinical manifestations of allergic rhinitis and asthma, though the strength of the correlation can vary depending on the allergen tested [23].

Despite its advantages, SPT has limitations. For instance, there can be a risk of false negatives, particularly in cases of recent antihistamine use or an improper technique [12]. Additionally, false positives can occur due to irritant reactions rather than true allergic sensitizations [12]. Consequently, while SPT remains a cornerstone in allergy diagnostics, especially in respiratory and food allergies, it is often used in conjunction with sIgE testing and a thorough patient history to confirm diagnoses and guide treatment plans [12]. These studies collectively underscore that while SPT is a valuable and widely used diagnostic tool, its results must be interpreted within the broader clinical context, with consideration given to the method’s limitations.

In vitro, total IgE testing measures the overall levels of IgE in the bloodstream and lacks specificity. On the other hand, specific IgE testing targets particular allergens, both food and inhalants, through blood sample collection from the individual [22]. While specific IgE testing provides high sensitivity and specificity, it requires a longer turnaround time, typically around 10–20 days, to obtain results [24].

Specific IgE (sIgE) testing is a widely used diagnostic tool in allergy testing and it is designed to detect IgE antibodies specific to particular allergens in a patient’s blood. Numerous studies have compared sIgE testing with other diagnostic methods such as skin prick testing (SPT) and component-resolved diagnostics (CRD). For example, Wood et al. found that sIgE testing is particularly sensitive in detecting sensitization to common allergens like pollen, house dust mites, and pet dander [25]. However, the study noted variability in specificity depending on the allergen and the chosen cutoff values [14]. Another study by Matricardi et al. compared sIgE testing to CRD and concluded that CRD offers a more refined identification of allergen sensitization, particularly in distinguishing primary sensitization from cross-reactivity, which is critical in complex allergy cases [26]. Additionally, a study by Zuberbier et al. underscored that while sIgE testing is a powerful tool for detecting allergic sensitizations, it should not be used in isolation due to the potential for false positives, particularly in individuals with high total IgE levels [27]. This analysis highlighted the importance of interpreting sIgE results within the broader context of a patient’s clinical history and other diagnostic tests. Ewan and Dugue also discussed the possibility of false positives in sIgE testing, especially in patients with elevated total IgE levels, further emphasizing the need for a comprehensive approach to allergy diagnosis [28]. These studies collectively suggest that while sIgE testing is valuable in allergy diagnostics, it should be integrated with other diagnostic methods to achieve the most accurate results.

##### Common Allergens and Their Prevalences According to Diagnosis Techniques

Food allergies are a growing concern globally, with significant variations in prevalence across different populations and age groups. Studies have consistently identified certain foods as the most common triggers of IgE-mediated allergic reactions, with cow milk, gluten, eggs, wheat, beans, soybean, nuts, and seafood being the primary culprits behind food allergies [1]. A landmark study by Sicherer and Sampson in the United States highlighted that peanut allergy, affecting approximately 1–2% of children, is the most prevalent food allergy in Western countries, with a rising incidence over the past two decades [29]. Another critical review by Gupta et al. explored the burden of food allergies across the globe, noting that while peanut and tree nut allergies are particularly prevalent in Western nations, rice and sesame are more common allergens in Asia and the Middle East [30]. Moreover, the Australian Health Nuts study emphasized the significant role of environmental and genetic factors in the development of food allergies, with peanut and egg allergies being the most common among Australian children, affecting nearly 10% of infants by the age of one [31]. These studies collectively underscore the importance of geographic and demographic factors in determining the prevalence and types of food allergies, and they highlight the need for region-specific public health strategies to manage and prevent food allergies effectively.

#### 3.1.4. Mixed IgE/Non-IgE-Mediated Food Allergies

##### Eosinophilic Esophagitis

Eosinophilic esophagitis (EoE) is a chronic, immune-mediated esophageal disease characterized by the infiltration of eosinophils in the esophageal epithelium, leading to symptoms such as dysphagia and food impaction [32]. The rising incidence of EoE has prompted numerous studies aimed at understanding its pathophysiology, prevalence, and treatment options. A pivotal study by Dellon et al. highlighted the increasing prevalence of EoE in both pediatric and adult populations in the United States, noting a significant rise in diagnoses over the past two decades [33]. This study underscored the role of environmental factors, such as food allergens and aeroallergens, in the pathogenesis of EoE [33].

Another key study by Liacouras et al. provided a comprehensive review of diagnostic criteria and treatment guidelines for EoE, emphasizing the importance of dietary management, including elimination diets and the use of proton pump inhibitors (PPIs), in managing the condition [34]. The study compared the effectiveness of different therapeutic approaches, including dietary elimination and pharmacologic treatments, concluding that while both strategies are effective, the choice of the treatment should be individualized based on patient characteristics and preferences [34].

Further research by Hirano et al. explored the long-term outcomes of patients with EoE, revealing that while the disease is chronic and relapsing, early diagnosis and consistent treatment can significantly improve quality of life and reduce the risk of complications such as esophageal strictures [35]. This study also compared the effectiveness of topical corticosteroids, such as fluticasone and budesonide, with dietary interventions, finding that while corticosteroids are effective in reducing inflammation, dietary changes are crucial in managing symptoms in the long term [35].

Finally, a meta-analysis by Lucendo et al. compared the outcomes of various treatments for EoE, including elimination diets, PPIs, and corticosteroids [36]. The study concluded that while elimination diets are effective in inducing histologic remission, PPIs and corticosteroids also play a vital role, particularly in patients who do not respond to dietary changes [36].

These studies collectively highlight the complex and multifactorial nature of EoE, emphasizing the need for individualized treatment plans that consider both dietary and pharmacologic interventions to manage this chronic condition effectively.

##### Non-IgE-Mediated Food Allergies

Non-IgE cell-mediated food allergies pose a greater diagnostic challenge and are categorized into various disorders, including allergic proctocolitis (AP), celiac disease/dermatitis herpetiformis, food protein-induced enteropathy (FPE), Heiner syndrome (pulmonary hemosiderosis), food protein-induced enterocolitis syndrome (FPIES), and cow’s milk (CM) protein-induced iron deficiency anemia [4,5,17].

These allergies are mostly diagnosed during early childhood, except for celiac disease, with the main allergens being identified as wheat and soy and cow milk [37].

##### Allergic Proctocolitis

Formerly recognized as allergic colitis, this benign condition primarily affects young children [37]. It is characterized by the presence of bright red blood in the stool (hematochezia), often accompanied by diarrhea, and is primarily linked to the consumption of cow and soy milk in children [10,11]. The diagnosis of allergic proctocolitis (AP) involves clinical evaluation, laboratory tests, stool examination (fecal calprotectin), endoscopic procedures, and allergy assessments (specific IgE and skin prick testing) [38,39]. It is marked by mild anemia, an elevated eosinophil count (eosinophilia), higher-than-normal total IgE levels, the presence of eczema, and hypoalbuminemia [10,11,12]. Treatment typically entails eliminating the offending food trigger, leading to symptom resolution within a maximum of ninety-six hours [37]. According to Elizur et al., the prevalence of AP in young children is 1.6 per 1000 infants [40].

Another review study performed by Mennini et al. found that 0.16% of healthy children and 64% of children suffering from blood in stools are in fact suffering from allergic proctocolitis [41].

Furthermore, Vassilopoulou et al. found similar results to Osborne et al. [31] regarding the prevalence of the disease. Cow’s milk (83%), eggs (7.3%), wheat (6.4%), and beef (6.4%) were identified as the main triggers of allergic proctocolitis symptoms, caused by the ingestion of these food proteins through the mother’s diet during breastfeeding [32].

A 2019 study by Nowak-Węgrzyn et al. provided a comprehensive overview of the condition, emphasizing that while allergic proctocolitis is typically benign and self-limited, occurring in infants within the first few months of life, it requires careful dietary management to avoid complications [42]. This study stressed the importance of eliminating the offending protein from the maternal diet in breastfeeding infants or switching to a hypoallergenic formula for formula-fed infants [7].

A more recent study by Ruffner et al. explored the long-term outcomes of infants diagnosed with allergic proctocolitis, revealing that most children outgrow the condition by the age of one, though a small percentage may develop other atopic conditions later in life. This study highlighted the need for ongoing monitoring, particularly in infants with a family history of atopic diseases [8].

Further research by Caubet et al. emphasized the role of maternal diet during pregnancy and breastfeeding in the prevention and management of allergic proctocolitis, suggesting that maternal dietary restrictions can significantly reduce the incidence and severity of symptoms in infants at risk [43]. This study provided evidence supporting the early identification and elimination of food allergens as a key strategy in managing allergic proctocolitis effectively [43].

These studies collectively highlight the evolving understanding of allergic proctocolitis, emphasizing the importance of personalized dietary management strategies, ongoing monitoring, and the potential benefits of maternal dietary modifications. They underline the significance of early diagnosis and intervention to prevent long-term complications and ensure optimal growth and development in affected infants.

##### Celiac Disease/Dermatitis Herpetiformis

Celiac disease (CD) is an immune-mediated disorder characterized by a permanent immune response that is triggered by consuming gluten-containing foods such as wheat, barley, and rye [44]. Classified as an enteropathy, it leads to the severe malabsorption of several vitamins (D, B12, B6), minerals (iron), and macronutrients due to the atrophy of intestinal villi, primarily in the duodenum, rendering individuals with CD susceptible to various nutritional deficiencies [15,16]. Most CD patients are reported to carry haplotype HLA DR3-DQ2 and/or DR4-DQ8, serving as susceptibility indicators in celiac disease prediction [45]. Endoscopic duodenal intestinal biopsy is considered the gold standard for diagnosing celiac disease [46]. Serologic testing, including tissue transglutaminase (TTG), endomysia antibodies (EMAs), deamidated gliadin peptide (DGP), and antigliadin antibodies (AGAs), can help in discovering CD, with AGAs being less specific and gradually replaced by newer tests like EMA, TTG, and DPG [47]. Treatment involves completely eliminating gluten from the diet for life, which resolves intestinal damage [47], and 1.4% of the global population is estimated to be suffering from celiac disease [48].

According to West et al., the incidence of celiac disease was estimated to be around 19.1 per 100,000, a figure that has been increasing since the early 2000s [14].

A 2020 study by Lebwohl et al. investigated the global prevalence of celiac disease, finding significant geographical variation, with higher rates in Europe and the United States compared to Asia and Africa [44]. This study underscored the role of genetic predisposition and environmental factors in the development of the disease [44].

In contrast, a 2019 study by Ludvigsson et al. focused on the diagnostic approaches, comparing serological tests such as anti-tissue transglutaminase (tTG) antibodies with biopsy findings [49]. The study emphasized the high sensitivity and specificity of tTG testing, making it a cornerstone in the non-invasive diagnosis of celiac disease, though biopsies remain the gold standard for confirmation [49].

A 2021 systematic review by Pinto-Sánchez et al. evaluated various dietary management strategies, particularly the effectiveness of a strict gluten-free diet (GFD) in achieving mucosal healing and reducing symptoms [9]. The review concluded that while most patients benefit from a GFD, there is a subset of patients with non-responsive celiac disease who may require additional interventions, such as the exclusion of trace gluten or refractory celiac disease therapies [9].

Further research by Rubio-Tapia et al. explored the long-term outcomes of patients with celiac disease, particularly focusing on the risks of complications such as enteropathy-associated T-cell lymphoma (EATL) [50]. This study highlighted the importance of early diagnosis and strict adherence to a GFD for reducing the risk of such severe complications [50].

These studies collectively contribute to a nuanced understanding of celiac disease, highlighting the importance of early and accurate diagnosis, effective management through a strict gluten-free diet, and ongoing monitoring to prevent complications and improve quality of life.

Similarly, dermatitis herpetiformis (DH) is an autoimmune skin condition characterized by the formation of small blisters or papules, rashes, and urticaria, typically appearing on the elbows, knees, and buttocks [51]. DH shares a similar genetic background to CD, with some researchers considering it a subtype of CD [52]. The incidence of DH is decreasing globally, while CD is on the rise, with DH being more prevalent in males and CD being more prevalent in females [18,20]. The diagnosis of DH primarily involves examining clinical signs and symptoms and confirming them through the direct immunofluorescence examination of perilesional skin, revealing granular immunoglobulin A (IgA) in the papillary dermis [18,20]. Patients with DH may also experience gastrointestinal symptoms and nutritional deficiencies related to CD, which resolve upon complete gluten avoidance [18]. Although adopting a gluten-free diet resolves urticaria and blisters, it may take months to years for skin symptoms to completely clear [18,20].

DH’s incidence is estimated at 0.8 per 100,000 people [21]. The prevalence is thought to be between 11.2 and 75.3 per 100,000 individuals [53]. Another study, conducted by Antiga et al. [54], showed the same prevalence number in the population while additionally showing that the disease is almost absent in African and Asian populations but occurs frequently in the Caucasian population. This is mainly due to the absence of the human leukocyte haplotypes (HLAs) DQ2 and DQ8 from the African and Asian populations and low wheat consumption in these regions [54].

Reunala et al. confirmed that DH is a specific manifestation of celiac disease, with nearly all patients exhibiting some degree of intestinal involvement, even if asymptomatic [55]. This study highlighted the importance of adhering to a strict gluten-free diet (GFD) as the primary treatment for DH, alleviating the skin symptoms and addressing the underlying intestinal inflammation associated with celiac disease [55].

Further research by Mansikka et al. provided insights into the epidemiology of DH, showing that its incidence has decreased over the past few decades, likely due to better recognition and earlier diagnosis of celiac disease [56]. This population-based study from Finland reported a decrease in the prevalence of DH, which the authors attributed to the widespread adoption of gluten-free diets among individuals with celiac disease and the earlier initiation of treatment before the development of skin symptoms [56].

Additionally, Collin et al. examined the long-term outcomes of DH patients, particularly focusing on the risk of associated autoimmune diseases and malignancies [52]. Their findings suggested that while DH patients on a strict GFD had a similar overall mortality risk to the general population, those who did not strictly adhere to the diet had an increased risk of developing other autoimmune conditions, particularly thyroid disease, and certain cancers such as non-Hodgkin lymphoma [52].

Moreover, a review by Caproni et al. summarized the immunopathogenesis of DH, emphasizing the role of IgA deposits in the skin and the cross-reactivity between epidermal transglutaminase (the autoantigen in DH) and tissue transglutaminase (the autoantigen in celiac disease) [57]. This understanding has informed the diagnostic approaches, with skin biopsies for direct immunofluorescence being the gold standard for diagnosing DH, and serological tests for IgA antibodies being used to support the diagnosis of celiac disease [57].

Celiac disease and DH are nonetheless diseases that can cause many serious health problems if not diagnosed early. Following a lifetime avoidance of gluten is currently the only solution to these diseases.

##### Food Protein-Induced Enteropathy (FPE)

Food protein-induced enteropathy (FPE) is a chronic, non-IgE-mediated gastrointestinal disorder that primarily affects infants. It is characterized by persistent diarrhea, malabsorption, and failure to thrive due to intolerance to specific dietary proteins, particularly cow’s milk and soy [58]. Comparative studies on FPE and related conditions, such as food protein-induced enterocolitis syndrome (FPIES), have highlighted distinct differences in clinical presentation and progression [58]. Jenkins et al. established cow’s milk as a predominant trigger in FPE, with symptoms typically emerging in the first few months of life [59]. The review of 462 patients with FPE provided insights into the variable prognosis, where some children outgrow the condition, while others continue to experience symptoms into later childhood [8]. Caubet et al. further underscored the challenge of managing FPE in cases of multiple food sensitivities, stressing the need for personalized dietary interventions [60]. Moreover, Fernandes et al. explored the potential for FPE to persist into adulthood, which remains a growing area of interest [61]. These studies suggest that while FPE shares certain clinical characteristics with other non-IgE-mediated disorders, its management requires a more nuanced approach, particularly in children with complex dietary protein intolerances. The evolving body of research advocates the continuous monitoring and reevaluation of dietary strategies as children age, ensuring both symptom control and optimal growth and development.

##### Heiner Syndrome (Pulmonary Hemosiderosis)

Cow milk has been identified as the primary trigger in Heiner syndrome (pulmonary hemosiderosis), a non-IgE-mediated allergy characterized by pulmonary diseases in young infants [62]. This condition is considered rare and often requires time for diagnosis [63]. It typically presents gastrointestinal symptoms, poor growth, anemia, pulmonary hemosiderosis (PH), and symptoms resembling pulmonary infections, which resolve after removing cow milk from the diet [26]. Diagnosis involves clinical evaluation, laboratory tests (cow milk IgE and IgG antibodies), and radiography [64].

Despite its rarity, Heiner syndrome remains an important differential diagnosis in infants with recurrent pulmonary infections and anemia, especially when symptoms do not respond to conventional treatments [65]. Misdiagnosis is common, as symptoms overlap with chronic lung diseases such as cystic fibrosis and recurrent aspiration pneumonia [64]. This highlights the need for increased awareness among pediatricians and pulmonologists to ensure timely diagnosis and intervention.

While cow milk elimination remains the primary treatment, studies suggest that delayed diagnosis can lead to long-term complications, including chronic lung damage and persistent iron deficiency anemia [63]. This raises concerns about whether early nutritional interventions, such as alternative hypoallergenic formulas or hydrolyzed protein-based diets, should be implemented in high-risk infants to prevent disease progression [66].

Emerging research also suggests that genetic and environmental factors may play a role in the development of Heiner syndrome. Some studies have hypothesized that certain genetic predispositions may influence immune responses to food proteins, leading to an exaggerated inflammatory reaction in the lungs [67]. However, further research is needed to confirm these associations and explore potential biomarkers that can be used for early detection.

Overall, while Heiner syndrome is still relatively under-researched, the available studies underscore the importance of awareness and early intervention to prevent long-term complications in affected children. Future studies should focus on identifying risk factors, improving diagnostic accuracy, and evaluating the effectiveness of novel therapeutic strategies, such as immunomodulatory treatments or microbiome-based interventions, to better manage the condition.

##### Food Protein-Induced Enterocolitis Syndrome (FPIES)

Food protein-induced enterocolitis syndrome (FPIES) represents another form of non-IgE-mediated food allergy, characterized by severe delayed gastrointestinal symptoms that typically occur within the first year of life. Symptoms include repeated vomiting, hypotension, blood in the stool, and diarrhea following the ingestion of the offending food [68]. The diagnosis of FPIES primarily relies on clinical symptoms, elimination diets, oral food challenges, and the assessment of symptom progression in patients [69]. Cow milk is the most common trigger food for this syndrome [17]. 0.34% is the current prevalence of the disease in pediatric patients up to 3 years of age, with 90% of patients recovering at the age of 3, with the disease is diagnosed in the majority of patients in the 6 first months of life [61].

In Australia, it is estimated that up to 90 percent of patients suffering from FPIES visit allergy clinics for their condition, with an estimation of 1 in 10,000 Australian infants less than 2 years of age having the disease [70].

A recent study by Ruffner et al. delved into the immune mechanisms underlying FPIES, proposing that T-cell responses may play a pivotal role in its pathogenesis, contrasting the traditional focus on innate immune responses [8]. These studies emphasize the need for tailored approaches to managing FPIES, including individualized elimination diets and careful food reintroduction protocols. Given the severity of FPIES reactions and the potential for misdiagnosis, there is a growing call for the development of standardized diagnostic criteria and improved awareness among healthcare providers [71]. Delayed diagnosis can lead to unnecessary medical interventions and prolonged dietary restrictions, affecting a child’s nutritional status and growth [72]. Moreover, research suggests that the gut microbiome may play a role in disease progression, with dysbiosis potentially influencing immune responses to trigger foods [73]. This raises questions about whether probiotic interventions could help modulate immune tolerance in FPIES patients, an area that warrants further investigation.

Furthermore, the variability in triggers and the immune mechanisms involved suggest that a one-size-fits-all approach may not be adequate.

Future research should focus on identifying potential biomarkers for early detection, as well as refining food challenge protocols to minimize risk while ensuring accurate diagnosis. Additionally, the psychological impact on parents and caregivers should not be overlooked, as managing FPIES requires strict vigilance in food selection and preparation, which can contribute to anxiety and stress.

### 3.2. Food Intolerances

Food intolerances, another facet of food hypersensitivity, refer to non-immunological reactions triggered by a food or food component typically tolerated in certain doses [74]. It is estimated that up to 20% of the global population experiences food intolerance [30]. However, diagnosing this condition often requires an understanding of various clinical presentations, including the intensity and timing of symptom onset. Complicating matters further are the diverse modes of action of food intolerance, which may include pharmacological effects (such as with coffee), enzyme deficiencies (like lactose malabsorption), and nonspecific gastrointestinal functioning [30]. Lactose, gluten/wheat, histamine-rich foods, and FODMAPs are among the most commonly implicated triggers for food intolerances [75]. Gastrointestinal symptoms like bloating, gas, diarrhea, abdominal pain, and nausea are typical presentations of food intolerance, with life-threatening reactions being rare [17].

#### 3.2.1. Lactose Intolerance

Lactose intolerance (LI) occurs when individuals with lactose malabsorption (LM) experience symptoms like diarrhea, bloating, nausea, and abdominal pain after consuming lactose-containing foods [76]. It is estimated to affect between 57 and 65 percent of the global population [77]. Lactose, a disaccharide sugar found in most dairy products, is broken down into glucose and galactose by the enzyme lactase [75]. LM is a prerequisite for LI and can stem from various causes, including primary lactase deficiency (a gradual decline in lactase levels as individuals age) [78], secondary lactase deficiency (resulting from intestinal epithelium injury due to conditions like AIDS, chemotherapy, or gastrointestinal infections, this issue is reversible upon treatment of the underlying cause) [79], congenital lactase deficiency (a rare pediatric genetic disorder characterized by severe symptoms and failure to thrive) [80], and developmental lactase deficiency (occurring in premature infants with immature gastrointestinal systems) [75]. Diagnosis of LM typically relies on non-invasive methods like hydrogen breath tests [81], although other approaches such as genetic testing and enzymatic assays exist, with enzymatic assay measurements via bowel biopsies considered the gold standard [38,44,46]. Treatment options for LM include avoiding lactose-containing foods, using oral lactase enzyme replacements, and incorporating probiotics like Lactobacillus spp., Bifidobacterium longum, or Bifidobacterium into the diet. These probiotics have been demonstrated to stimulate the production of lactase when consumed [75].

A pivotal study by Swallow identified genetic polymorphisms associated with lactase persistence in populations of European descent, shedding light on the evolutionary aspects of lactose tolerance [82]. The study demonstrated that the ability to digest lactose into adulthood is primarily due to genetic mutations that allow for the persistence of lactase production beyond infancy [82]. This genetic advantage likely provided a nutritional benefit in pastoral societies, where dairy was a major food source [82].

Further exploring dietary management, a study by Shaukat et al. systematically reviewed the efficacy of different interventions for lactose intolerance [83]. The review found that while lactase enzyme supplements can help reduce symptoms, many individuals benefit from gradually introducing small amounts of dairy into their diet to build tolerance [83]. This approach, known as lactose adaptation, takes advantage of the fact that some individuals with lactose intolerance can tolerate up to 12 g of lactose (the amount in one cup of milk) without significant symptoms [83].

The variation in symptoms and tolerance levels underscores the need for individualized dietary recommendations based on genetic background and symptom severity.

#### 3.2.2. Non-Celiac Gluten/Wheat Sensitivity (NCGWS)

Non-celiac gluten/wheat sensitivity (NCGWS) refers to the condition experienced by individuals sensitive to gluten or wheat but lacking immune serological celiac antibodies or allergic biomarkers, with a higher prevalence observed among females [84]. The Salerno Experts’ Criteria, relying on exclusion diagnosis, is currently the only reliable diagnostic tool for NCGWS [5]. Manifestations of NCGWS include both intestinal (bloating, diarrhea, constipation, nausea, etc.) and non-intestinal/extraintestinal symptoms (headache, anxiety, weight loss, anemia) [85]. NCGWS shares some similarities and differences with wheat allergy and celiac disease, as summarized in Table 3 [84]. However, the exact mechanism underlying NCGWS remains poorly understood, leading some individuals to be prescribed a gluten-free diet by healthcare practitioners, despite negative results on other tests for gluten-containing foods [5].

Several studies have aimed to define the characteristics, prevalence, and management of NCGWS, often highlighting the complexity and controversies surrounding its diagnosis. A 2015 study by Catassi et al. reviewed the diagnostic criteria and noted that while NCGWS shares symptoms with celiac disease, such as bloating, diarrhea, and fatigue, it lacks the serological markers and histological changes seen in celiac disease [5]. The study also highlighted the placebo effect in gluten challenge trials, which complicates the diagnosis further, suggesting that some cases of NCGWS might be due to non-gluten components like FODMAPs [5].

In contrast, a 2016 study by Uhde et al. explored the immunological response in NCGWS patients and found evidence of systemic immune activation in response to gluten, without the intestinal damage typical of celiac disease [87]. This study suggested that NCGWS might involve an innate immune response rather than the adaptive immune response seen in celiac disease [87]. However, the study also acknowledged the heterogeneity of the condition, indicating that different patients might react to different components of wheat, including gluten or other proteins [87].

Further research by Skodje et al. compared the effects of gluten, FODMAPs, and placebos in NCGWS patients and concluded that many individuals who believed they were sensitive to gluten were reacting to FODMAPs, which are poorly absorbed carbohydrates found in wheat and other foods [88]. This study emphasized the importance of using a structured dietary approach to diagnose NCGWS, as misdiagnosis can lead to unnecessary dietary restrictions and nutritional deficiencies [88].

These studies highlight the complexities in diagnosing and managing NCGWS, emphasizing that while some individuals may indeed react to gluten, others might be sensitive to other components in wheat or even experience a placebo effect. The variability in immune responses and symptoms underscores the need for personalized approaches in both diagnosis and treatment [89,90].

We assessed non-celiac gluten/wheat sensitivity (NCGWS), IgA anti-EMA (IgA antibodies against endomysium), IgA anti-tTG (IgA antibodies against transglutaminase), and IgG anti-DGP (IgG antibodies against deamidated gliadin peptides).

#### 3.2.3. Fructose Intolerance

##### Hereditary Fructose Intolerance

Hereditary fructose intolerance (HFI) is a rare autosomal hereditary disorder characterized by the inability to metabolize fructose directly or indirectly through sucrose or sorbitol, as noted by Singh and Sarma in 2022 [91]. Individuals with fructose intolerance may experience symptoms such as abdominal pain, diarrhea, nausea, and flatulence upon consuming fructose-rich foods like honey, fruits, or vegetables [92]. The mutation of Aldolase B, the primary enzyme responsible for fructose metabolism in the liver, located on chromosome 9q22.3, is implicated in HFI [91]. The diagnosis of HFI typically involves the Benedicts test, performing the glucose dipstick test in urine, and serum carbohydrate-deficient transferrin (CDT). These are complemented by clinical correlations, and sometimes liver biopsies are required to assess Aldolase B enzyme activity [93]. The treatment of HFI primarily revolves around adopting a diet that is low in fructose, sucrose, and sorbitol (FSS). In acute cases, patients may require admission to an intensive care unit for intravenous glucose administration to manage metabolic acidosis, and it is crucial for individuals with HFI to avoid medications and vaccines containing sucrose, such as the rotavirus oral vaccine [91].

A study by Ali et al. identified several mutations in the *ALDOB* gene responsible for HFI, including the common A149P mutation, which is prevalent in European populations [94]. This research provided insight into the genetic basis of HFI and established the importance of genetic testing for accurate diagnosis [94].

In a more recent study, Tolan reviewed the pathophysiology of HFI and discussed the clinical symptoms that arise due to the accumulation of toxic metabolites, such as fructose-1-phosphate, in the liver, kidneys, and intestines [95]. This study highlighted the importance of early diagnosis and strict dietary management to avoid severe complications like hypoglycemia, liver dysfunction, and failure to thrive in infants [95]. Tolan also discussed the role of genetic counseling for families with a history of HFI and the potential for prenatal diagnosis [95].

These studies underline the genetic complexity of HFI and the critical importance of early diagnosis and dietary management in preventing serious health issues. The advancements in genetic testing have greatly improved the accuracy of HFI diagnosis, enabling better patient outcomes.

##### Non-Hereditary Fructose Intolerance

Non-hereditary fructose intolerance, or fructose malabsorption (FM), presents as a syndrome where the uptake of fructose in the small intestine is minimal in some individuals, leading to the fermentation of unabsorbed fructose in the colon, akin to lactose intolerance and HFI, according to [96]. The diagnosis of FM typically involves a specifically designed hydrogen breath test [97]. Treatment options primarily involve the elimination of fructose-containing foods from the diet, with xylose isomerase proposed as an oral treatment for converting fructose into glucose, resulting in favorable hydrogen breath test results in FM patients [98].

A study by Gibson et al. investigated the role of the low-FODMAP diet in managing fructose malabsorption [99]. This study highlighted the fact that fructose, along with other fermentable carbohydrates, contributes to symptoms like bloating and diarrhea in susceptible individuals [99]. The low-FODMAP diet was shown to be effective in reducing these symptoms, indicating that dietary management is a critical component of treatment for non-hereditary fructose intolerance [99].

Further research by Tuck et al. focused on the diagnostic challenges associated with fructose malabsorption [74]. The study noted that breath hydrogen testing is commonly used to diagnose this condition, but the results can be inconsistent due to variations in individual gut microbiota and other factors [74]. Tuck et al. advocated for a more comprehensive approach to diagnosis, combining dietary history with symptom tracking and possibly genetic testing, although the latter is more relevant to distinguishing between HFI and fructose malabsorption [74].

These studies suggest that non-hereditary fructose intolerance is a complex and multifaceted condition that requires careful dietary management and accurate diagnosis. Unlike HFI, which is caused by a genetic mutation, fructose malabsorption often arises from dietary factors and gut microbiota composition, making individualized treatment plans essential.

#### 3.2.4. Saccharose Intolerance

Saccharose intolerance arises from deficiencies in sucrase-isomaltase enzyme function, caused by congenital sucrase-isomaltase deficiency (CSID) or secondary factors such as celiac disease or Crohn’s disease [98]. Symptoms typically involve gastrointestinal issues such as cramps, bloating, gas, and diarrhea [100]. The diagnosis of saccharose intolerance often involves duodenal endoscopic biopsies or breath tests (hydrogen, C-sucrose) [54]. Treatment may include using sacrosidase enzyme supplementation, which has shown promising results in alleviating symptoms [54].

A 2020 study by Treem et al., published in the *Journal of Pediatric Gastroenterology and Nutrition*, highlighted the various genetic mutations affecting the SI gene and discussed enzyme replacement therapies like sacrosidase as an effective treatment [101]. The study also pointed out the importance of dietary modifications, emphasizing the use of low-sucrose and low-starch diets to manage the symptoms effectively [88]. Furthermore, a study by Robayo-Torres et al. highlighted the importance of genetic screening and enzyme activity testing in diagnosing CSID, emphasizing that many cases remain undiagnosed due to the overlap of symptoms with other gastrointestinal disorders like irritable bowel syndrome (IBS) [102]. Misdiagnosis remains a challenge, as individuals with CSID often experience symptoms similar to functional gastrointestinal disorders, leading to unnecessary dietary restrictions or ineffective treatments [103]. This underscores the need for heightened clinical awareness and improved diagnostic protocols, particularly in patients with persistent unexplained gastrointestinal symptoms [104]. This study also pointed to enzyme replacement therapy, particularly with sacrosidase, as an effective treatment option for managing symptoms, alongside dietary modifications that limit sucrose and starch intake [86]. However, long-term adherence to dietary changes and enzyme therapy requires patient education and continuous monitoring, as variations in food processing and enzyme activity levels can affect symptom control [105]. Future research should explore individualized treatment approaches, including personalized enzyme dosing and probiotic interventions, to optimize therapeutic outcomes [106].

#### 3.2.5. Histamine Intolerance

Histamine intolerance (HIT) refers to a non-immunological condition believed to result from elevated histamine levels in the blood due to the ingestion of histamine-rich foods. It can potentially cause adverse effects [107]. Histamine intoxication, on the other hand, occurs following the ingestion of histamine-rich foods, with symptoms ranging from gastrointestinal symptoms to skin reactions, low blood pressure, headaches, and palpitations [108]. Suspected causes of histamine intolerance include a lack of oxidative degradation via exposure to diamine oxidase (DAO) activity or reduced levels of methylation due to exposure to histamine *N*-methyltransferase (HNMT) [109]. This deficiency may be congenital or acquired, with factors such as gastrointestinal disorders (e.g., inflammatory bowel disease, celiac disease), chronic alcohol consumption, and certain medications (e.g., NSAIDs, histamine-releasing drugs) contributing to reduced DAO activity [108]. Studies suggest that DAO activity is predominantly localized in the intestinal mucosa, meaning that any disruption in gut health, such as increased intestinal permeability or dysbiosis, may exacerbate symptoms in susceptible individuals [11]. Common symptoms of HIT predominantly affect the gastrointestinal tract, with constipation, diarrhea, abdominal pain, and postprandial fullness being most prevalent [110]. However, the variability in symptoms between individuals suggests that other factors, such as genetic predisposition, gut microbiota composition, and cumulative histamine load from endogenous and exogenous sources, may play a role in symptom severity [11]. This variability also makes HIT difficult to diagnose, as symptoms overlap with those of other functional gastrointestinal disorders like irritable bowel syndrome (IBS) and non-celiac gluten sensitivity [111]. Treatment for HIT typically involves adhering to a low-histamine diet, with antihistamines used to alleviate symptoms [112]. However, managing dietary intake can be challenging due to the fluctuating histamine content in foods, storage conditions, and individual tolerance thresholds [113]. Fermented foods, aged cheeses, cured meats, and alcohol are well-documented triggers, yet some individuals may tolerate small amounts without symptoms, complicating dietary recommendations [113]. Mast cell stabilizers may also be utilized, although further research is needed to determine their efficacy in HIT treatment [114]. Oral supplementation with exogenous DAO has shown promise in reducing symptoms, but additional studies with larger sample sizes are required to ascertain its effectiveness in HIT patients [115].

Histamine intolerance (HI) is an emerging area of research. Various studies exist examining its prevalence, pathophysiology, and management strategies and exploring the physiological mechanisms of histamine metabolism and its relation to HIT [116]. They emphasize that histamine intolerance results from an imbalance between histamine release and its degradation due to enzyme deficiencies, particularly a diamine oxidase (DAO) deficiency. Another significant contribution is made by Schink et al., who provide insights into diagnostic approaches and the clinical management of HIT [11]. They suggested that HIT can be managed through dietary modifications and the use of DAO supplements. These studies underscore the multifaceted nature of HIT and highlight the importance of a comprehensive approach in both diagnosis and management [11]. Histamine intolerance remains an under-researched condition, with ongoing debates regarding its true prevalence, diagnostic criteria, and pathophysiological mechanisms. While histamine skin prick tests, plasma histamine levels, and DAO activity measurements have been suggested as potential diagnostic tools, none have been universally accepted due to inconsistencies in sensitivity and specificity [117]. Given the overlap of symptoms with other conditions, a comprehensive diagnostic approach—including elimination diets, symptom tracking, and laboratory assessments—is crucial for accurately identifying HIT. Future research should focus on refining diagnostic criteria, understanding individual variations in histamine metabolism, and exploring novel treatment approaches, such as targeted enzyme therapy or gut microbiota modulation, to improve patient outcomes [118].

#### 3.2.6. FODMAP

FODMAPs, or Fermentable Oligo-, Di-, Mono-Saccharides, and Polyols, are short-chain carbohydrates found abundantly in fruits, vegetables, dairy, cereals, and sweeteners [99]. Lactose, fructose, sorbitol, mannitol, fructans, stachyose, and raffinose are all classified as FODMAPs [119]. The consumption of more than 4 g of lactose, or more than 0.3 g of mannitol, sorbitol, galacto-oligosaccharides, or fructans, is considered to constitute a high-FODMAP diet [120]. FODMAPs are poorly digested by intestinal bacteria, leading to the production of short-chain fatty acids (SCFAs) and gas, which can cause symptoms such as abdominal pain, flatulence, diarrhea, and indigestion in susceptible individuals, particularly those with irritable bowel syndrome (IBS) [121]. A low-FODMAP diet (LFD) has been shown to effectively reduce symptoms and improve the quality of life in IBS patients [120]. The LFD typically follows a three-phase approach [12]. Phase 1 involves eliminating all FODMAPs from the diet for 4 to 6 weeks. Phase 2 assesses each patient’s tolerance to FODMAP subgroups by reintroducing one food at a time over three days. Phase 3, building upon the findings of Phase 2, customizes a long-term FODMAP diet tailored to each patient’s tolerance [122,123]. However, concerns have been raised regarding potential nutritional deficiencies, constipation, eating disorders, and alterations in gut microbiota among individuals on a long-term LFD [104,105,106,124].

A seminal paper by Gibson et al. in *Gastroenterology* introduced the low-FODMAP diet and demonstrated its effectiveness in reducing IBS symptoms through a controlled trial [99]. Their research provided the foundation for subsequent studies exploring dietary interventions [99]. Another comprehensive review by Halmos et al. supports the use of the low-FODMAP diet, showing that it significantly alleviates IBS symptoms compared to a standard diet [12]. This study is crucial for understanding the clinical application of the diet [12]. Additionally, a more recent study by Staudacher et al. confirms the diet’s effectiveness across various populations, emphasizing its role in symptom management [125,126]. These studies emphasize the low-FODMAP diet’s role in managing IBS and other functional gastrointestinal disorders, providing robust evidence for its clinical use.

### 3.3. Irritable Bowel Syndrome IBS

Irritable bowel syndrome (IBS) affects approximately 20% of the global population [127]. Its primary symptoms include abdominal pain, bloating, and irregularities in stool form and frequency [128]. Currently, there are no globally recognized biomarkers for diagnosing IBS; instead, diagnostic criteria based on symptoms have been established by the scientific community, known as the Rome IV criteria [129].

Patients are categorized into subtypes based on the Bristol stool form: IBS with constipation (IBS-C), IBS with diarrhea (IBS-D), IBS with mixed stool pattern (IBS-M), and IBS unclassified (IBS-U) [73,74]. IBS is a complex syndrome influenced by multiple factors and it often overlaps with various comorbidities, even among its own subtypes [130,131].

Common gastrointestinal disorders that overlap with IBS include gastroesophageal reflux (GERD), nausea, constipation, diarrhea, heartburn, dyspepsia, and incontinence [77,78]. Non-gastrointestinal syndromes also frequently overlap with IBS [73,74]. These include psychiatric conditions such as depression, anxiety, and somatization, as well as premenstrual syndrome (PMS), an overactive bladder, fibromyalgia, eating disorders, and food hypersensitivities (intolerances and allergies) [132,133].

This overlap has prompted discussion within the scientific community about whether IBS should be considered part of these syndromes rather than a distinct syndrome on its own [133]. Additionally, IBS and its associated functional gastrointestinal and non-gastrointestinal disorders are now classified under the somatic symptom disorders in the *Diagnostic and Statistical Manual of Mental Disorders, 5th Edition* (DSM-5). This categorization reflects a historical trend where patients with gastrointestinal symptoms treated by psychiatrists often experienced inadequate treatment and care [134].

Numerous risk factors associated with the development of IBS have been identified, including personal factors such as being female and having a low body mass index (BMI) [135].

Psychological factors like anxiety, depression, and low quality of life also increase somatic issues such as diverticulosis, antibiotic use, gastrointestinal infections, and endometriosis, as well as social conditions like a family history of mental illness, childhood socioeconomic status, and marital status [135].

Moreover, many individuals diagnosed with IBS also report food allergies or intolerances, particularly to gluten, dairy (cow milk protein and lactose), and FODMAPs, which can trigger adverse reactions [49,136,137].

The pathophysiology of IBS involves various mechanisms. These include alterations in gut microbiota, changes in the epithelial barrier, immune system responses to food antigens and bile acids, and interactions within the brain–gut axis, enteric nervous system, and hypothalamus–pituitary–adrenal axis, which are increasingly considered potential biomarkers of IBS [138].

Additionally, psychological factors such as depression, anxiety, and stress contribute to the pathophysiology of IBS by influencing intestinal motility [138].

As mentioned previously, the composition of gut microbiota in patients with IBS varies depending on the subtype of the condition, as detailed in Table [138]. Studies have demonstrated that culturing these microbiota types present in IBS patients, compared to those in individuals without IBS, can lead to lower intestinal motility, induce visceral hypersensitivity, and alter transit time. These findings on microbiota dysbiosis have been recognized by the ROME foundation [138].

One pivotal study by Longstreth et al. provides a comprehensive review of IBS epidemiology, diagnostic criteria, and management options, emphasizing the heterogeneity of the disorder and the importance of a tailored treatment approach [139]. Another influential study by Ford et al. examines the efficacy of pharmacological treatments for IBS, including antispasmodics, laxatives, and antidiarrheals, and finds that while some medications are effective, their benefits vary among patients [10]. A more recent randomized controlled trial by Drossman et al. investigates the effectiveness of a novel treatment approach, such as a combination of dietary interventions and pharmacological agents, and showed significant improvement in IBS symptoms compared to traditional therapies [140]. Additionally, a meta-analysis, provided an overview of the impact of the low-FODMAP diet on IBS symptoms, confirming its efficacy in symptom relief and highlighting the need for personalized dietary strategies [128]. Furthermore, a review study performed by Pasta et al. provides an in-depth analysis of the relationship between irritable bowel syndrome (IBS), food allergies, and food intolerances, highlighting their overlapping symptoms and distinct mechanisms. It emphasizes that while food intolerances, particularly fermentable carbohydrates (FODMAPs) and lactose, are common in IBS, true food allergies involving immune system activation are rare in IBS patients. The study underscores that many IBS symptoms, such as bloating, abdominal pain, and altered bowel habits, are frequently mistaken for allergic reactions, leading to unnecessary dietary restrictions. The review also discusses the importance of using accurate diagnostic methods to differentiate between IBS-related food intolerances and immune-mediated allergies, ensuring effective dietary management and symptom relief [141]. Lastly, a review by Quigley et al. explored the role of probiotics and prebiotics in managing IBS, revealing that while some evidence supports their use, more research is needed to establish their effectiveness conclusively [142]. Together, these studies underscored the complexity of IBS and the need for a multifaceted approach to its management.

## 4. Conclusions

Based on the current review, understanding the key differences between food allergies and intolerances is crucial for effective management and treatment strategies. While both conditions can lead to uncomfortable symptoms, they arise from distinct mechanisms within the body.

Food allergies involve an immune system response to specific proteins in food, often resulting in rapid and potentially life-threatening reactions. On the other hand, food intolerance stems from difficulty digesting certain foods or components, leading to gastrointestinal discomfort or other symptoms. These reactions are typically less severe and do not involve the immune system.

Recognizing the symptoms and triggers of each condition is essential for accurate diagnosis and appropriate management. Food allergies may require the strict avoidance of triggering foods and the availability of emergency medication, such as epinephrine, in cases of severe reactions. Meanwhile, managing food intolerances often involves identifying and eliminating problematic foods from the diet, as well as considering enzyme supplements and other supportive measures.

Overall, by understanding the nuances of food allergies and intolerances, individuals can take proactive steps to minimize their symptoms and maintain their overall health and well-being. Consulting with healthcare professionals and registered dietitians can provide personalized guidance and support for navigating these dietary challenges.

## Figures and Tables

**Figure 1 nutrients-17-01359-f001:**
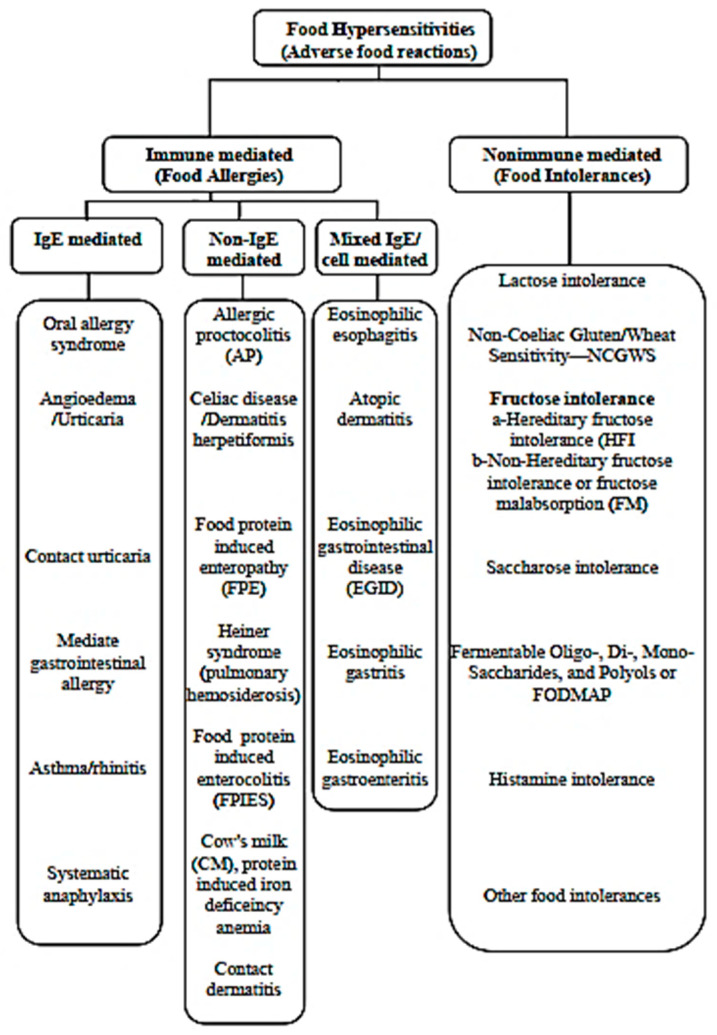
Food hypersensitivities: classification and subclassification infographic [3,4].

**Figure 2 nutrients-17-01359-f002:**
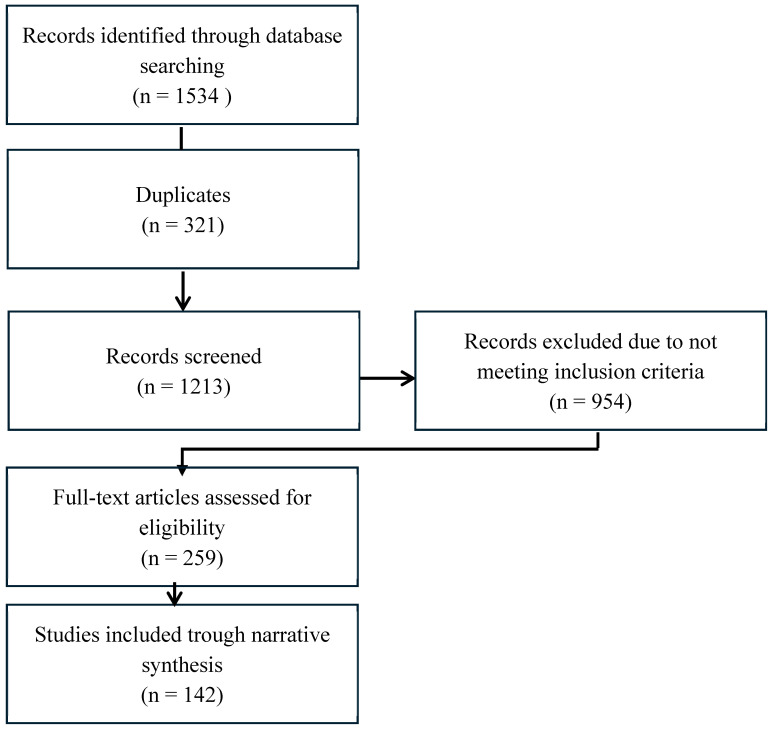
Flowchart of study selection.

**Table 1 nutrients-17-01359-t001:** Main characteristics of included studies.

Author	Year	Study Type	Country	Sample Size	Duration of Intervention	Summary of Findings
Vassilopoulou et al. [7]	2022	Retrospective, observational, multicenter case–control study	Greece	96 mothers of infants with and 141 mothers of infants without a history of Food Protein-Induced Allergic Proctocolitis (FPIAP).	From May 2018 to November 2020	Identified cow milk (83%), eggs (7.3%), wheat (6.4%), and beef (6.4%) as the main triggers for allergic proctocolitis (AP) in infants through the maternal diet.
Ruffner et al. [8]	2013	Retrospective chart review	USA	462 cases were identified from the hospital patients	From 2007 until 2012	Food protein-induced enterocolitis syndrome (FPIES) reactions were observed more frequently than previously reported, though their presentation and clinical characteristics remained consistent with earlier findings. Milk- and soy-induced FPIES were prevalent, with 43.5% of patients who reacted to milk also experiencing a reaction to soy.
Pinto-Sánchez et al. [9]	2021	Prospective study	Canada	Prospective study of 50 patients with irritable bowel syndrome (IBS) (ROME III, all subtypes), with and without serologic reactivity to gluten (antigliadin IgG and IgA), and 25 healthy subjects (controls)	Between 2012 and 2016	Evaluated the effectiveness of a gluten-free diet in achieving mucosal healing for celiac patients.
Ford et al. [10]	2014	Cross-sectional	Canada	4224 patients recruited	Between January 2008 and December 2014	Functional bowel disorders (FBDs) showed significant demographic and psychological differences among patients. The Rome III classification system did not clearly distinguish between different FBD subtypes. There was considerable symptom overlap among irritable bowel syndrome (IBS), functional diarrhea, and chronic idiopathic constipation (CIC). The findings suggest a need for improved diagnostic criteria to differentiate FBDs more effectively.
Schink et al. [11]	2018	Cross-sectional observational study	Germany	64 participants 8 with histamine intolerance (HIT), 25 with food hypersensitivity (FH), 21 with food allergy and 10 healthy controls (HC)	12 months	Suggested dietary modifications and DAO supplements for histamine intolerance.
Halmos et al. [12]	2014	Randomized, controlled, cross-over trial	Australia	30 patients with IBS and 8 healthy individuals (controls, matched for demographics and diet)	Between April 2009 and June 2011	Confirmed the efficacy of the low-FODMAP diet in IBS symptom reduction.
Nwaru et al. [13]	2014	Systematic review and meta-analysis	Europe	Not applicable	Between 1 January 2000 and 30 September 2012	Highlighted that early introduction of allergenic foods may reduce the risk of developing IgE-mediated food allergies.
West et al. [14]	2014	Observational population-based study	UK	57 million	Between 1990 and 2011	Found that the incidence of celiac disease is increasing, estimating 19.1 per 100,000 cases annually.

**Table 2 nutrients-17-01359-t002:** Food allergy disorders and main features.

Pathology	Disorder	Key Features	Most Common Causal Foods
IgE-mediated (acute onset)	Acute urticaria/angioedema	Food commonly causes acute (20%) but rarely chronic urticaria	Cow milk, gluten, eggs, wheat, beans, soybean, nuts, and seafood
	Contact urticaria	Direct skin contact results in lesions. Histamine release, in rare cases, can cause urticaria.	Multiple
	Anaphylaxis	Rapidly progressive, multiple organs system reactions can include cardiovascular collapse	Any but more commonly peanut, tree nuts, shellfish, fish, milk, and egg
	Food-associated, exercise-induced anaphylaxis	Food only triggers anaphylaxis if ingestion is followed temporally by exercise	Wheat, shellfish, and celery are most often described
	Oral allergy syndrome (pollen-associated food allergy syndrome)	Pruritus and mild edema are confined to the oral cavity and uncommonly progress beyond the mouth (w7%), and rarely to anaphylaxis (1% to 2%). Its incidence might increase after the pollen season.	Raw fruit/vegetables; cooked forms tolerated; examples of relationships: birch (apple, peach, pear, carrot), ragweed (melons)
	Immediate gastrointestinal hypersensitivity	Immediate vomiting, pain	Cow milk, gluten, eggs, wheat, beans, soybean, nuts, and seafood
Combined IgE and cell-mediated (delayed onset/chronic)	Atopic dermatitis	Associated with food allergy in 35% of children with moderate-to-severe rash	Major allergens, particularly egg, milk
	Eosinophilic esophagitis	Symptoms might include feeding disorders, reflux symptoms, vomiting, dysphagia, and food impaction.	Multiple
	Eosinophilic gastroenteritis	Vary on site(s)/degree of eosinophilic inflammation; might include ascites, weight loss, edema, obstruction	Multiple
Cell-mediated (delayed onset/chronic)	Food protein-induced enterocolitis syndrome cow’s milk, soy, rice, oat, meat	Primarily affects infants. Chronic exposure: emesis, diarrhea, poor growth, lethargy. Re-exposure after restriction: emesis, diarrhea, hypotension (15%) 2 h after ingestion.	Cow’s milk, soy, rice, oat, meat
	Food protein induced allergic proctocolitis	Mucus-laden, bloody stools in infants Milk (through breast-feeding)	Milk (through breast-feeding)
	Allergic contact dermatitis	Often occupational because of chemical moieties, oleoresins. Systemic contact dermatitis is a rare variant because of ingestion	Spices, fruits, vegetables
	Heiner syndrome	Pulmonary infiltrates, failure to thrive, iron deficiency anemia	Cow’s milk

**Table 3 nutrients-17-01359-t003:** Comparison of prevalence, pathogenic, and diagnostic features of gluten-related disorders.

	Celiac Disease	NCGWS	Wheat Allergy
Prevalence	0.5–1.7%	0.6–10%	0.5–9% in children
Pathogenesis	Autoimmune	Non-specific immune response	IgE mediated response
DQ2-DQ8 HLA haplotypes	Positive in 95% cases	Positive in 50% cases	Negative
Serological markers	IgA anti-EMA, IgA anti-tTG, IgG anti-DGP, IgA anti-gliadin	IgA/IgG anti-gliadin in 50% cases	specific IgE antibodies against wheat and gliadin
Duodenal biopsy *	Marsh I to IV with domination of Marsh III and IV	Marsh 0-II, but according to some experts Marsh III might also be in NCGS	Marsh 0-II
Duodenal villi atrophy	Present	Absent	Might be present or absent

* Marsh classification (histological grading system used to evaluate and classify the degree of intestinal damage, particularly in the small intestine, in individuals with celiac disease or other gluten-sensitive enteropathies [86]).

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
