# Peer review of "Food Hypersensitivity: Distinguishing Allergy from Intolerance, Main Characteristics, and Symptoms—A Narrative Review"

_nutrients, 2025, doi:10.3390/nu17081359_

Round 1

Reviewer 1 Report

Comments and Suggestions for Authors

This overview fulfills its informative role and has practical value. It is well edited and can be published.  Nevertheless, I suggest the following additions:

  1. The executive summary contains only general information and makes little reference to the content of the review.
  2. The introduction should include the reasons and purpose of the research undertaken.
  3. Chapter 3.3. It contains some simplifications regarding the pathogenesis of IBS, including the role of the gut microbiome. I propose to remove Table 4, because Ref. 127 is one of over 1000 publications on this problem, with still heterogeneous results and conclusions. The content of the chapter may remain unchanged, except for the information that SCFAs have a pro-inflammatory effect, as their effect is bidirectional and depends on their concentration.

In addition, I propose  to include  REF.138 – Pasta A. et al. Food  Intolerance, Food Allergies and IBS: Lights and Shadows.  Nutrients, 2024, 16,265 – .

Author Response

"We would like to express our sincere thanks to the reviewer for their detailed and thoughtful comments on our manuscript. We appreciate the time and effort put into reviewing our work.

Reviewer’s comment: The executive summary contains only general information and makes little reference to the content of the review. The introduction should include the reasons and purpose of the research undertaken.

Author’s Reply:  The executive summary has been updated to incorporate more information about the article with the purpose also added of this research. (please refer from line 19 to 27).

Reviewer’s comment: Chapter 3.3. It contains some simplifications regarding the pathogenesis of IBS, including the role of the gut microbiome. I propose to remove Table 4, because Ref. 127 is one of over 1000 publications on this problem, with still heterogeneous results and conclusions. The content of the chapter may remain unchanged, except for the information that SCFAs have a pro-inflammatory effect, as their effect is bidirectional and depends on their concentration.

Author’s Reply:  In chapter 3.3 table 4 has been removed and the paragraph about SCFAs also has been removed.

Reviewer’s comment: REF.138 – Pasta A. et al. Food  Intolerance, Food Allergies and IBS: Lights and Shadows.  Nutrients, 2024, 16,265 –

Author’s Reply: This reference has been added as suggested by the reviewer, and placed at the end of the IBS section just before the conclusion. After reviewing the article, we found it to be an important addition to our discussion and relevant to the scope of our review. (please refer to lines 803 to 813.

Once again, we thank the reviewers for their constructive feedback and hope that the revisions meet the expectations. We believe the manuscript has been significantly improved as a result of these valuable suggestions.

Author's Reply to the Review Report (Reviewer 2)

"We would like to express our sincere thanks to the reviewer for their detailed and thoughtful comments on our manuscript. We appreciate the time and effort put into reviewing our work.

Reviewer’s comment: The manuscript presents an interesting review devoted to food hypersensitivity, allergy, and intolerance. The article brings interesting information but is not focused. The information is mainly basic, more appropriate for a student textbook or a more basic journal than reviewing recent developments in the field or highlighting chosen aspects of the problem.

Author’s Reply: Thank you very much for your thoughtful feedback. We appreciate your insight and understand your concern about the perceived broad scope of the review. Our intent with this article was to provide a comprehensive overview of food hypersensitivity, allergy, and intolerance, including basic concepts, to establish a solid foundation for readers who may not be familiar with the topic. We aim to create an accessible resource that introduces key aspects of the field while also addressing both foundational and emerging concepts.

However, we understand the importance of focusing on recent developments and highlighting specific aspects of the problem in a review for a more specialized audience. In light of your feedback, we have revised the manuscript according to the 4 reviewers’ comments and suggestions and made some modifications to the content of this review article to achieve the purpose and aim that we are trying to share with the scientific community.

We hope that these revisions better align the manuscript with the expectations for an in-depth review while retaining the comprehensive approach that we initially envisioned.

Thank you again for your constructive comments, which have helped us improve the focus and relevance of the manuscript. (please refer to the text written in red for all the modifications made throughout this article; (pages 430-453, 472-480, 483-487, 655-659, 661-666, 675-681, 683-694,709-717)

Reviewer’s comment: The limitation of the search to Pubmed might have omitted papers published in food journals not covered by this database.

Author’s Reply: Thank you for pointing out the potential limitation of restricting the search to PubMed. We acknowledge that this approach may have excluded relevant studies published in food-specific journals that are not indexed by PubMed.

In addition to PubMed, we also conducted a thorough search using Google Scholar to capture a broader range of articles, including those that might be found in food-related journals. However, we did not explicitly mention Google Scholar in the manuscript. We regret this omission and will now include it in the methods section to ensure transparency regarding the search strategy. (please refer to line 63)

We believe this dual search approach helped us cover a broad spectrum of research, but we appreciate your feedback and revised the manuscript accordingly to clarify the search strategy.

Thank you once again for your valuable suggestion.

Reviewer’s comment: Table 1: FPIES, FPIAP, IBS, Please define the acronyms used (even if they may be obvious for specialists)

Author ‘s Reply:  Thank you for your suggestion. The relative acronyms were defined in our updated manuscript. (please refer to table 1 page 4 to 6)

Reviewer’s comment: Table 1: “I change the sentence) (nonimmunologic)”, unclear. Should it be removed?

Author’s Reply:  Thank you for suggesting the removal of this phrase. It was initially added for correction but was unintentionally left in during the proofreading process. We have now removed it in the updated manuscript. (please refer to table 1 page 4 to 6)

Once again, we thank the reviewers for their constructive feedback and hope that the revisions meet the expectations. We believe the manuscript has been significantly improved as a result of these valuable suggestions.

Reviewer 2 Report

Comments and Suggestions for Authors

The manuscript presents an interesting review devoted to food hypersensitivity, allergy and intolerance. The article brings interesting information but is not focused. The information is mainly basic, more appropriate for a student textbook or a more basic journal than reviewing recent developments in the field or highlighting chosen aspects of the problem.

Limitation of the search to Pubmed might have omitted papers published in food journals not covered by this database.

Table 1: FPIES, FPIAP, IBS, Please define the acronyms used (even if they may be obvious for specialists)

Table 1: “I change the sentence) (nonimmunologic)”, unclear. Should it be removed?

Literature should be uniformly formatted according to the requirements of the journal.

Author Response

We would like to express our sincere thanks to the reviewer for their detailed and thoughtful comments on our manuscript. We appreciate the time and effort put into reviewing our work.

Reviewer’s comment: The manuscript presents an interesting review devoted to food hypersensitivity, allergy, and intolerance. The article brings interesting information but is not focused. The information is mainly basic, more appropriate for a student textbook or a more basic journal than reviewing recent developments in the field or highlighting chosen aspects of the problem.

Author’s Reply: Thank you very much for your thoughtful feedback. We appreciate your insight and understand your concern about the perceived broad scope of the review. Our intent with this article was to provide a comprehensive overview of food hypersensitivity, allergy, and intolerance, including basic concepts, to establish a solid foundation for readers who may not be familiar with the topic. We aim to create an accessible resource that introduces key aspects of the field while also addressing both foundational and emerging concepts.

However, we understand the importance of focusing on recent developments and highlighting specific aspects of the problem in a review for a more specialized audience. In light of your feedback, we have revised the manuscript according to the 4 reviewers’ comments and suggestions and made some modifications to the content of this review article to achieve the purpose and aim that we are trying to share with the scientific community.

We hope that these revisions better align the manuscript with the expectations for an in-depth review while retaining the comprehensive approach that we initially envisioned.

Thank you again for your constructive comments, which have helped us improve the focus and relevance of the manuscript. (please refer to the text written in red for all the modifications made throughout this article; (pages 430-453, 472-480, 483-487, 655-659, 661-666, 675-681, 683-694,709-717)

Reviewer’s comment: The limitation of the search to Pubmed might have omitted papers published in food journals not covered by this database.

Author’s Reply: Thank you for pointing out the potential limitation of restricting the search to PubMed. We acknowledge that this approach may have excluded relevant studies published in food-specific journals that are not indexed by PubMed.

In addition to PubMed, we also conducted a thorough search using Google Scholar to capture a broader range of articles, including those that might be found in food-related journals. However, we did not explicitly mention Google Scholar in the manuscript. We regret this omission and will now include it in the methods section to ensure transparency regarding the search strategy. (please refer to line 63)

We believe this dual search approach helped us cover a broad spectrum of research, but we appreciate your feedback and revised the manuscript accordingly to clarify the search strategy.

Thank you once again for your valuable suggestion.

Reviewer’s comment: Table 1: FPIES, FPIAP, IBS, Please define the acronyms used (even if they may be obvious for specialists)

Author ‘s Reply:  Thank you for your suggestion. The relative acronyms were defined in our updated manuscript. (please refer to table 1 page 4 to 6)

Reviewer’s comment: Table 1: “I change the sentence) (nonimmunologic)”, unclear. Should it be removed?

Author’s Reply:  Thank you for suggesting the removal of this phrase. It was initially added for correction but was unintentionally left in during the proofreading process. We have now removed it in the updated manuscript. (please refer to table 1 page 4 to 6)

Once again, we thank the reviewers for their constructive feedback and hope that the revisions meet the expectations. We believe the manuscript has been significantly improved as a result of these valuable suggestions.

Reviewer 3 Report

Comments and Suggestions for Authors

This review aims to present a classification of the diseases caused by food hypersensitivity, as well as mechanisms and clinical picture.

Both figure 1 and figure 2 have low quality, regarding design and clarity. I do not know if this is what you submitted or the electronic submission system lowered the quality of the figures. Anyhow, improvments must be made. 

In figure 1 under non-Ige... you cite a disease (Allergic proctopilis (AP)). Did you mean allergic
proctocolitis? The lines between classes and diseases are also unclear and need better resolution.

Figure 2 is largely unreadable. It is very unclear how you ended up with 8 papers in table 1. As I understand you screened 1223 articles and rejected 1397 (440 because of inclusion criteria and 957 because of methods and topics)? It is not logic... And then you talk about 137 studies included through narrative synthesis... Are they included in those 1223 or not? Please include in your article a paragraph where the relation between all these numbers is clear so that we can get an explanation regarding the reason you chose the papers in table 1.

Another problem with the 8 articles that you chose in table 1 is that they do not cover all the diseases that you talk about in this systematic review, according to what you describe i column "summary of findings". A more careful presentation of literature selection is mandatory.

Author Response

"We would like to express our sincere thanks to the reviewer for their detailed and thoughtful comments on our manuscript. We appreciate the time and effort put into reviewing our work.

Reviewer’s comment: Both Figure 1 and Figure 2 have low quality, regarding design and clarity. I do not know if this is what you submitted or if the electronic submission system lowered the quality of the figures. Anyhow, improvements must be made.

Author’s Reply:  Thank you for pointing out the issues with the quality of Figures 1 and 2. We apologize for the low design and clarity, which may have been affected during the submission process. We have now improved the quality of both figures, enhancing their design and clarity. The updated figures have been included in the revised manuscript, and we believe they now meet the required standards. (please refer to figures 1, and 2, pages 2 and 3).

Reviewer’s comment: In Figure 1 under non-Ige... you cite a disease (Allergic proctopilis (AP)). Did you mean allergic

proctocolitis? The lines between classes and diseases are also unclear and need better resolution.

Author’s Reply:  Thank you for bringing this to our attention. The disease mentioned should be "allergic proctocolitis" rather than "allergic proctopilis." We have corrected this in the revised figure. This error was caused by the autocorrect feature of Microsoft word between the French language and the English language. The autocorrect feature was also rectified in Microsoft word.

Additionally, we appreciate your observation regarding the clarity of the lines between classes and diseases. We have improved the resolution of the figure and made adjustments to enhance the distinction between these categories for better clarity. (please refer to figure 1, page 2)

Thank you again for your helpful feedback, which has contributed to improving the quality of the figure

Reviewer’s comment: Figure 2 is largely unreadable. It is very unclear how you ended up with 8 papers in table 1. As I understand you screened 1223 articles and rejected 1397 (440 because of inclusion criteria and 957 because of methods and topics)? It is not logic... And then you talk about 137 studies included through narrative synthesis... Are they included in those 1223 or not? Please include in your article a paragraph where the relation between all these numbers is clear so that we can get an explanation regarding the reason you chose the papers in table 1.

Author’s Reply:  Thank you for your valuable feedback. We understand the confusion regarding the study selection numbers, and we appreciate the opportunity to clarify this.

Upon re-evaluating our flowchart, we realized that there was an error in the calculations, which led to the confusion. We sincerely appreciate your attention to detail, as this allows us to improve the clarity and accuracy of our work.

To correct this, we have reviewed the numbers carefully and adjusted them to ensure logical consistency

We initially identified 1,534 records through database searches.

After removing 321 duplicates, 1,213 records remained for screening.

Of these, 954 records were excluded for not meeting inclusion criteria, leaving 259 full-text articles for eligibility assessment. (please refer to page 3, figure 2)

From these 259 full-text articles, 152 studies were included in the final narrative synthesis. (An additional reference was added as suggested by reviewer number 1). (other references were added as suggested by reviewer 2)

Once again, we thank the reviewers for their constructive feedback and hope that the revisions meet the expectations. We believe the manuscript has been significantly improved as a result of these valuable suggestions.

Reviewer 4 Report

Comments and Suggestions for Authors

I have general and specific comments

First the general:

  1. This systematic review is taking on too many things.  A systematic review, esp. with a topic this rich, should have a focused research question.  Then the review should address everything related to that.  For example, focusing only on casein

The specific:

Title… symptoms: a systematic review.

Avoid 1st person language throughout

L23: big differences is not quantifiable

The introduction sees very short and I worry that you are biting off too much

For a systematic review it focused research question is ideal.  It looks like you are looking at everything.  Why include other review papers?

Is Figure 2 a PRISMA?  That should be described in the methods

Was this registered with PROSPERO?  That is standard

My recommendation is to 1) focus on a clinical question, 2) register your SR with Prospero, 3) revise your paper around that one question

Author Response

"We would like to express our sincere thanks to the reviewer for their detailed and thoughtful comments on our manuscript. We appreciate the time and effort put into reviewing our work.

Reviewer’s comment: This systematic review is taking on too many things.  A systematic review, esp. with a topic this rich, should have a focused research question.  Then the review should address everything related to that.  For example, focusing only on casein.

Author’s Reply: Thank you for your comments. The main objective of the study was to differentiate between food allergies and food intolerances according to symptoms, complications, and treatments.  Speaking about one single item that induces allergic reactions or intolerance will change all the objectives of the article

Reviewer’s comment: Title… symptoms: a systematic review.

Author’s Reply: This has been changed (title in red)

Reviewer’s comment: Avoid 1st person language throughout

Author’s Reply: This has been adjusted in the article.

Reviewer’s comment: L23: big differences are not quantifiable

Author’s Reply: This has been changed

Reviewer’s comment: The introduction is very short and I worry that you are biting off too much.

Author’s Reply: The introduction has been expanded (line 43-51).

Reviewer’s comment: For a systematic review it focused research question is ideal.  It looks like you are looking at everything.  Why include other review papers?

Author’s Reply: The research questions are: What are the key characteristics of food hypersensitivities? How do food allergies and food intolerances differ in terms of characteristics, treatment, and symptoms?

These systematic reviews necessitated many review articles.

Reviewer’s comment: Is Figure 2 a PRISMA?  That should be described in the methods

Author’s Reply: This has been changed (line 74- 77)

Reviewer’s comment: Was this registered with PROSPERO?  That is standard.

Author’s Reply: This review has been registered with Prospero

Once again, we thank the reviewers for their constructive feedback and hope that the revisions meet the expectations. We believe the manuscript has been significantly improved as a result of these valuable suggestions.

Round 2

Reviewer 2 Report

Comments and Suggestions for Authors

I accept the explanations and amendments

Author Response

I accept the explanations and amendments

Author's Reply 

"We would like to express our sincere thanks to the reviewer for his detailed and thoughtful comments on our manuscript. We appreciate the time and effort put into reviewing our work.

Reviewer 3 Report

Comments and Suggestions for Authors

The article is acceptable in its present form.

Author Response

The article is acceptable in its present form.

Author's answers

We would like to express our sincere thanks to the reviewer for his detailed and thoughtful comments on our manuscript. We appreciate the time and effort put into reviewing our work

Reviewer 4 Report

Comments and Suggestions for Authors

I would like to commend the authors on their additions to their text.

The major issue I have is making this a systematic review.  A definition I found online: 

A systematic review is a type of literature review that uses systematic and explicit methods to identify, select, critically appraise, and synthesize all relevant research on a specific topic or question. The goal is to provide a comprehensive, unbiased, and evidence-based summary of the existing knowledge.

Here's a breakdown of the key characteristics of a systematic review:

  • Clearly Defined Research Question: A systematic review focuses on a specific and well-formulated question, often using the PICO (Population, Intervention, Comparison, Outcome) framework

This paper should have a clear and focused question.  Instead, it appears to take on everything related to food allergies and intolerances.  Its reads more like a book chapter than a systematic review.

What I would have liked to seen was "does treatment A work better than treatment B for food allergy X".  You can define your PICO from that and then evaluate all literature related to that topic, ideally RCT.

Also, if this was a systematic review, it should also be registered with PROSPERO

Author Response

he major issue I have is making this a systematic review.  A definition I found online: 

systematic review is a type of literature review that uses systematic and explicit methods to identify, select, critically appraise, and synthesize all relevant research on a specific topic or question. The goal is to provide a comprehensive, unbiased, and evidence-based summary of the existing knowledge.

Here's a breakdown of the key characteristics of a systematic review:

  • Clearly Defined Research Question: A systematic review focuses on a specific and well-formulated question, often using the PICO (Population, Intervention, Comparison, Outcome) framework

Comments of Reviewer 4

This paper should have a clear and focused question.  Instead, it appears to take on everything related to food allergies and intolerances.  It reads more like a book chapter than a systematic review.

What I would have liked to seen was "does treatment A work better than treatment B for food allergy X".  You can define your PICO from that and then evaluate all literature related to that topic, ideally RCT.

Also, if this was a systematic review, it should also be registered with PROSPERO

Author's answer

Thank you for your feedback and for evaluating our revised manuscript.

We understand and appreciate your concerns regarding the reproducibility of the search strategy and the PROSPERO registration. While we strived to conduct a thorough and structured review, we acknowledge that certain elements typically required for a systematic review - such as a clearly stated Boolean search string and PROSPERO registration number—were not explicitly presented.

Therefore, we fully accept your recommendation to reclassify the manuscript as a narrative review and make the necessary adjustments to the title and throughout the text accordingly. (please review Lines 54, 77-78 for the changes, and in Figure 2 a reference about Prisma was removed )

Thank you again for your constructive guidance.